# Exploring the Aroma Profile of Traditional Sparkling Wines: A Review on Yeast Selection in Second Fermentation, Aging, Closures, and Analytical Strategies

**DOI:** 10.3390/molecules30132825

**Published:** 2025-06-30

**Authors:** Sara Sofia Pinheiro, Francisco Campos, Maria João Cabrita, Marco Gomes da Silva

**Affiliations:** 1LAQV/REQUIMTE, Department of Chemistry, NOVA School of Science and Technology, NOVA University Lisbon, 2829–516 Caparica, Portugal; ss.pinheiro@campus.fct.unl.pt; 2Amorim Cork, R&D + i Department, 4535–387 Santa Maria de Lamas, Portugal; 3MED, Mediterranean Institute for Agriculture, Environment and Development & CHANGE, Global Change and Sustainability Institute, Departamento de Fitotecnia, Escola de Ciências e Tecnologia, Universidade de Évora, Polo da Mitra, Ap. 94, 7006–554 Évora, Portugal; mjbc@uevora.pt

**Keywords:** sparkling wine, yeast selection, tirage closures, aging on lees, analytical methods

## Abstract

Sparkling wine is a complex alcoholic beverage with high economic value, produced through a secondary fermentation of a still wine, followed by a prolonged aging period that may last from nine months to several years. With the growing global demand for high-quality sparkling wines, understanding the biochemical mechanisms related to aroma development has become increasingly relevant. This review provides a comprehensive overview of the secondary fermentation process, with particular emphasis on yeast selection, types of closure, and the impact of aging on the volatile composition. Special attention is also given to the analytical strategies employed for the identification and quantification of target compounds in sparkling wine matrices. Due to the presence of volatile compounds at trace levels, effective extraction and pre-concentration techniques are essential. Extraction methods such as solid-phase microextraction (SPME), stir-bar sorptive extraction (SBSE), and thin-film SPME (TF-SPME) are discussed, as well as chromatographic techniques, such as gas chromatography (GC) and liquid chromatography (LC).

## 1. Introduction

According to The International Organisation of Vine and Wine (OIV), sparkling wines are considered a category of special wines and are characterized by the production of effervescence, resulting from the release of carbon dioxide (CO_2_) of endogenous origin [1].

Over the last few decades, the global wine market has experienced a significant rise in the demand for sparkling wines, driven by evolving consumer preferences and a growing appreciation for product quality. Although the production volume of sparkling wines remains lower than that of still wines, their economic impact on the oenology industry is significant due to their high economic added value [2,3]. According to OIV, between 2002 and 2018, the global sparkling wine market experienced a 57% increase in overall growth, reflecting its expanding role in both established and emerging markets [4]. In 2024, sparkling wines accounted for 10.9% of total exported volume and 23.8% of exported value [5].

The secondary fermentation of sparkling wines can be conducted via the traditional method or the Charmat/Tank method, which has given rise to sparkling wines with different characteristics. In the traditional method, formerly known as *Méthode Champenoise*, the base wine is refermented inside the bottle, followed by a long aging period in contact with yeast lees. This extended time allows the maturation of sparkling wine, producing complex aromas such as yeasty and toasted bread notes. On the other hand, in the Charmat method, the secondary fermentation is carried out in pressurized stainless-steel tanks (autoclaves) over a shorter period. As a result, the wines typically retain more primary aromas, such as fresh fruit and floral notes [3,6].

The traditional method is the most widely technique for high-quality sparkling wine production and gives rise to some of the most renowned sparkling wines in the world such as Champagne (France), Cava (Spain), Franciacorta (Italy), Sekt (Germany), and Cap Classique (South Africa) [4,6]. The success of this method has led to its adoption in various wine-producing regions such as the United Kingdom, Portugal, Brazil, and Australia, contributing to the expansion and diversification of the global sparkling wine industry [7]. As shown in Figure 1, Europe (Spain, France, Italy, and Germany) led the sparkling wine production in 2018, followed by Asia and America [4].

The quality of bottle-fermented sparkling wine is shaped by multiple factors, including grape variety, yeast strain selection, tirage conditions, and lees aging duration. Given its complexity, the traditional method has been the focus of extensive research, with studies critically analyzing its impact on wine composition and quality at each stage of production [7,8,9,10].

This review aims to present studies conducted on the selection of yeast strains and their influence on the aromatic profile of sparkling wines, the choice of closure at the time of tirage and its impact during the aging process, the aging process itself and the resulting changes in the aromatic profile, and the most commonly used methods for extraction, pre-concentration, and analysis for the identification and quantification of target analytes in sparkling wines.

In contrast to previous reviews, this work provides an integrated perspective by including the potential benefit of using cork closures during the tirage phase of sparkling wine production, along with a focus on both sample preparation and analytical techniques, aspects that are often overlooked in the literature.

## 2. Secondary Fermentation of Sparkling Wines

The sparkling wine production according to the traditional method (as shown in Figure 2) involves two sequential fermentations—in the first, the grape must is transformed into base wine, while the second, initiated by the addition of tirage liquor (a mixture of sucrose, yeast, nutrients, and a clarifying agent), is carried out in the bottle. Throughout the second fermentation, carbon dioxide (CO_2_) is formed, creating the effervescent characteristic of sparkling wine with internal pressures of approximately 5–7 atm, at 20 degrees Celsius [6]. Subsequently, the wine is stored horizontally while in contact with yeast less, for at least 9 months, depending on country regulations [8]. During this time, yeast autolysis takes place, and the final characteristics of the sparkling wine are obtained. Afterwards, the sparkling wine undergoes riddling, which allows the yeast sediment to accumulate in the neck of the bottle, followed by its removal through disgorgement.

### 2.1. Yeast Strain Selection

The sensory profile of sparkling wines made by the traditional method is strongly influenced by yeast selection for the secondary fermentation, and research suggests that different strains can significantly impact the aging process and final chemical composition [11]. Advances in yeast selection have facilitated the identification of strains specifically suited to winemaking, enhancing fermentation efficiency, stress resistance, and sensory quality [9,12].

The selection of yeast strains for refermentation is primarily guided by essential technological characteristics, such as tolerance to high pressure and alcohol content, as well as the ability to thrive at low temperatures with minimal production of SO_2_ and off-flavors. Additionally, the yeast strain should exhibit optimal flocculation and autolysis capacity during aging, thereby extending the interaction between the base wine and lees [6,13]. For this reason, a wide variety of *Saccharomyces cerevisiae* strains are commercialized and used for the first and second fermentation of grape musts and base wines, respectively [14]. Di Gianvito, P. et al. [15] tested six flocculent *Saccharomyces cerevisiae* wine strains with different flocculation degrees and autolytic activity and two commercial strains in traditional sparkling winemaking. The results showed a considerable diversification quantitatively and qualitatively in the production of aroma molecules (namely ethyl octanoate, responsible for sour and apple aroma, 3-methyl-1-butanol, and 2-phenylethanol, which contribute to herbaceous, rose, and sweet aroma), and sparkling wines obtained with autochthonous flocculent strains presented a higher number of alcohols and esters, already after 3 months, highlighting the possibility of exploring them as starter cultures to produce differentiated traditional sparkling wines [15].

A comprehensive review regarding biotechnologies in sparkling wine was published in 2011 by Torresi S. et al. [9] and it summarizes some *Saccharomyces cerevisiae* strains that produce good quality sparkling wines although with different characteristics, such as alcohol tolerance (*S. cerevisiae* (*ex bayanus*) BCS103), good flocculation capacity (*S. cerevisiae ph.r. bayanus* PB2019), the enhancement of sensory characteristics (*S*. *cerevisiae* (hybrid)), and autolytic capacity (*S*. *cerevisiae* r.f. *bayanus*) [9].

The use of commercial active dry yeast (ADY) is essential in the refermentation process of the traditional method for their ability to withstand high acidity, ethanol concentration, and low pH, and for their flocculation capacity under carbon dioxide pressure [8,16]. A study performed by Benucci I. et al. [16] on yeast dynamics during starter culture preparation (also known as *pied de cuve*) demonstrated that the kinetics of sugar consumption are initially rapid during acclimatization to the alcoholic medium but slow during active growth, regardless of the base wine tested. While initial differences in viability and population dynamics among ADY strains tend to diminish over time, specific strains exhibit distinct fermentative capacities. For instance, *Saccharomyces cerevisiae bayanus* R *Vitilevure* DV10 showed the highest sugar consumption rate during *pied de cuve*, whereas *S. cerevisiae bayanus Lalvin* EC-1118 achieved the greatest pressure increase during the second fermentation [16]. Furthermore, Berbegal, C. et al. [17] reported that Cava sparkling wines fermented with ADY exhibited significantly higher aroma scores, underscoring the impact of yeast selection on the sensory profile of the final product. These outcomes emphasize the role of commercial ADY, ensuring an efficient second fermentation and enhancing the organoleptic quality of sparkling wines [17].

In traditional sparkling wine production, riddling and disgorgement are demanding and time-intensive processes. To enhance efficiency and accelerate this step, the use of immobilized yeasts has also been investigated. Although their application in oenological practices remains relatively limited, a study by Lopéz de Lerma, N. et al. [18] on the influence of immobilized yeast strains on the aromatic profile of long-aged Spanish sparkling wines established that immobilized systems could serve as an effective alternative to free yeast cells, potentially improving aromatic quality [18]. On the other hand, Fernandez-Fernandez, E. et al. [19] found no significant differences in the aromatic profile between Verdejo wines fermented with free and immobilized yeast strains [19]. Despite this inconsistency, the use of immobilized yeasts in the second fermentation of sparkling wines may simplify riddling and disgorging processes, without negatively impacting key quality parameters.

More recently, the use of indigenous yeast strains as starter cultures, particularly for secondary fermentation, has gained attention as a strategy to enhance the distinctive characteristics of regional wines, as these strains are adapted to specific environmental conditions, contribute to organoleptic differentiation, and help mitigate fermentative issues [20,21,22]. Vigentini, I. et al. [23] screened 133 *S. cerevisiae* strains based on technological (fermentative power and vigor, SO_2_ tolerance, alcohol tolerance, and flocculence) and qualitative criteria (acetic acid, glycerol, and H_2_S production), finding that indigenous strains performed comparably to commercial starters. Their use in refermentation offers a viable approach to enhance product differentiation while preserving traditional winemaking techniques [23]. In 2018, Garofalo, C. et al. [24] confirmed the suitability of autochthonous *S. cerevisiae* to improve the quality of regional Apulia sparkling wines [24]. In agreement with this investigation, Tufariello, M. et al. [25], using Bombino Bianco and Nero grapes, showed that, regarding aroma and a metabolomic approach, autochthonous yeast strains can be an influential tool for innovation and market differentiation [25].

The growing economic interest in the sparkling wine sector has led to renewed focus on microbial resource management, particularly the use of non-*Saccharomyces* yeast, creating opportunities to produce innovative wines with unique characteristics to meet the demands of the competitive international market [12].

The key oenological properties of non-*Saccharomyces* yeasts have been extensively reviewed [26], along with their potential to address various technological and safety challenges, including the regulation of volatile acidity, alcohol reduction, increased glycerol production, enhancement of varietal aroma expression, and mitigation of contaminants.

The current knowledge of non-*Saccharomyces* yeasts in starter cultures for secondary fermentation has been reported, highlighting the impact on chemical composition and sensory attributes of traditional sparkling wine. The review was focused on *Torulaspora delbrueckii*, *Metschnikowia pulcherrima*, *Schizosaccharomyces pombe*, and *Saccharomycodes ludwigii*, with variations in amino acids, biogenic amines, volatile organic compounds (VOCs), aroma compounds, glycerol, and proteins described, which influence flavor and foaming properties. Due to limited studies on sparkling wines, findings from still wine have been used to assume potential effects, particularly concerning nitrogenous compounds, VOCs, proteins, organic acids, and sensory characteristics [12,27].

Ivit, N. et al. [27] explored making natural sparkling wines using non-*Saccharomyces* yeasts. Spanish base wine made from Airén (white) and Tempranillo (red) grapes was used in this experiment. Even though the total amount of volatile compounds was similar between yeast strains, *Schizosaccharomyces pombe* showed aromatic differences compared with the traditional *S. cerevisiae* and produced red sparkling wines with higher pyranoanthocyanin content and color intensity [27]. Another study demonstrated that the combination of killer and sensitive *S. cerevisiae* and *S. bayanus* strains can influence the concentration of polysaccharides, free amino acids, and total protein in a three-month-aged sparkling wine [28].

In a more innovative field, La Gatta, B. et al. [29] tested the addition of lees recovered from the first fermentation in tirage liquor and its influence on the sensorial profile of Bombino sparkling wines. Even though proteolysis increased, resulting in a positive effect on the aroma profile, the results showed a decrease in foam stability [29].

The use of specific non-*Saccharomyces* strains in conjunction with *S. cerevisiae* can positively modulate key chemical parameters and enhance the aromatic intensity of sparkling wine [30,31]. In this context, exploring yeast biodiversity emerges as a strategic approach to further optimizing production and improving sparkling wine quality.

### 2.2. Influence of Bottle Closure During Tirage

The second fermentation, known as *prise de mousse*, occurs after adding tirage liquor, a mixture of yeast, sucrose, nutrients, and, occasionally, bentonite. The wine is then sealed in bottles under controlled conditions, using either crown caps or cork closures, to enable secondary fermentation and subsequent aging.

Currently, crown caps are the preferred closure due to their ability to withstand high internal pressures, their compatibility with automated bottling and disgorgement processes, and lower prices [32,33]. However, in several wine regions, some producers continue the tradition of using cork stoppers during the second fermentation, believing it positively influences the sparkling wine’s sensory attributes [34]. Moreover, the use of cork stoppers aligns with environmental conservation and socio-economic sustainability, contributing to forest preservation and the prevention of soil degradation.

A study conducted by the Agricultural Research Council in South Africa investigated the influence on the sensory and aromatic quality of bottle-fermented sparkling wines (*Méthode Cap Classique*) sealed with cork stoppers or crown caps during the second fermentation and aging on lees. The first study [32] examined the effect of cork closures on the phenolic profile, analyzing the migration of gallic, caftaric, caffeic, and *p*-coumaric acids from three different cork types (Cork A, Cork R, and Cork C) and comparing them to wines sealed with crown caps. The results indicated that different cork types released varying levels of phenolic compounds, likely due to differences in surface roughness and contact area with the wine, with Cork A contributing the highest concentration of gallic acid. The second study [33] explored organoleptic differences between cork-closed and crown-capped sparkling wines, analyzing bottle pressure, polyphenol profiles, sensory attributes, and CO_2_ kinetics. Cork-sealed wines exhibited lower pressure yet remained within legal limits and demonstrated distinct polyphenol profiles. Sensory evaluation revealed that cork-closed wines had smaller bubbles and a longer aftertaste and retained CO_2_ longer after pouring compared to their crown-capped counterparts.

More recently, Jové, P. et al. [35] investigated the influence of different closures on the volatile composition of Gramona sparkling wine during second fermentation and bottle aging over a 94-month period. By analyzing six closure types, including cork stoppers and screw caps, using headspace-solid-phase microextraction (HS-SPME) and thermal desorption followed by gas chromatography coupled to mass spectrometry tandem (TD-GC-MS/MS), the study revealed that esters were the most abundant volatile compounds, with ethyl hexanoate (fruity and green apple aromas) predominating in screw caps and ethyl octanoate (floral and sweet notes) being more prevalent in cork stoppers. Alcohols such as isoamyl alcohol (banana notes) and phenylethyl alcohol (rose-like aromas) were present in both closures, while 1-butanol (medicinal aroma) was exclusive to screw caps. Additionally, the study identified closure-specific compounds, with screw caps containing aziridinylethylamine and hydroxyurea, whereas cork stoppers exhibited longifolenaldehyde and 6,7-dimethoxy-1,4-dimethyl-1,3-quinoxalinedithione, potentially contributing to woody and earthy aromas. The presence of dimethylamine in screw-cap wines suggested a possible impact on aged wine perception. Variability was also observed within closure subtypes, highlighting differences between agglomerated cork stoppers of varying diameters and between different screw-cap materials [35].

From a cork-focused perspective, the elemental composition of white sparkling wine and its respective cork stopper was studied over 18 months [36]. The cork was analyzed in three sections—external, bulk, and bottom layers—revealing that elements like Si, Ti, Fe, Ni, and Zn were concentrated in the outer layers, likely due to resin treatments, while Ba was associated with adhesive use. Over time, group II elements such as Mg, S, K, Ca, Cu, Sr, and especially Ba accumulated in the bottom layer, suggesting a migration process, with wine moisture acting as an easing medium. In the sparkling wine itself, most elemental concentrations increased except for Mg and Si, influenced by both cork interaction and yeast autolysis during secondary fermentation. Notably, Ba was absent in the sparkling wine, indicating its retention within the cork [36].

Although the current literature on sparkling wines and tirage cork stoppers is limited, the data presented in this paper support the idea that cork closures during the second fermentation and aging process can influence the evolution, style, chemical profile, and overall quality of bottle-fermented sparkling wines.

### 2.3. Autolysis and Aging on Lees

After the second fermentation, sparkling wines undergo aging in contact with lees. This process is considered one of the most important in defining the wine’s quality, as yeast autolysis induces chemical changes that contribute to the characteristic bouquet of sparkling wine. Contact with yeast lees also appears to protect the wine from oxidation and helps to prevent browning [13]. The duration of lees aging is regulated by national legislation, varying across production regions; for example, Cava requires a minimum of nine months, while Champagne demands at least twelve months [9]. Portuguese sparkling wines are primarily produced using techniques developed in the Champagne region, with an aging period of at least twelve months [37].

In the traditional method, prolonged lees aging is associated with improved organoleptic properties, contributing to aromatic complexity, structural balance, and longevity. Yeast autolysis involves the enzymatic degradation of cellular components, releasing mannoproteins, peptides, and ribonucleotides, which interact with wine constituents. The progressive breakdown of glucans and proteins enhances mouthfeel, providing roundness and texture, while increasing antioxidant capacity and color stability. Additionally, the release of ribonucleotides may intensify flavorful sensations [38].

Hervé, A. contributed to scientific discussions on yeast autolysis in sparkling wine through his involvement in the review article Yeast Autolysis in Sparkling Wine and a book chapter in Yeasts in the Production of Wine [39,40]. His work provides a detailed examination of the biochemical and morphological mechanisms underlying yeast autolysis, including enzymatic processes (such as the action of proteases and glucanases), cellular changes, and factors influencing autolysis. Additionally, the author presents a descriptive analysis of the findings reported over the years regarding the release of polysaccharides, lipids, nucleic acids, and volatile compounds during sparkling wine aging.

Sparkling wines aged in contact with lees are characterized by having toasty, lactic, sweet, and yeasty aromas. Throughout this period, compounds such as norisoprenoids, acetals, diacetyl, and furans tend to appear or increase over time. The overall ester concentration decreases, which may be attributed to their volatility, chemical hydrolysis, and possible adsorption onto yeast lees. Nevertheless, some esters, including diethyl succinate, ethyl lactate, and ethyl isovalerate, are described as aging markers due to their increase over time. Vitispirane was proposed as a marker for young sparkling wines [41,42,43]. Martín-García, A. et al. [44] identified over 60 volatile compounds in Cava sparkling wines influenced by aging conditions, with furans showing a strong correlation with time, suggesting their potential as aging markers [44].

To understand the changes in sparkling aroma with aging, Escudero A. et al. [45] studied the evolution of aroma in Champagne wines through normal and accelerated aging (by increasing temperature). Although common reactions occur in both aging methods, the aroma profiles were not identical. The results showed a decrease in the intensity of floral and fruity notes during natural aging and an increase in cooked aromas. The rise in temperature led to the formation of cis-3-hexenol from fatty acids and a decrease in some VOCs such as furaneol [45].

The impact of biological aging duration on nitrogen’s composition and the influence on sensory attributes of sparkling wine was also investigated [46]. The study, using non-traditional grape varieties (Niagara, Manzoni Bianco, Vilenave, Goethe, and Chardonnay) demonstrated that free amino acids increased over time, particularly citrulline, lysine, phenylalanine, glycine, aspartic acid, arginine, tyrosine, valine, and methionine, indicating their release from yeast cell walls. Principal component analysis (PCA) distinguished sparkling wines aged for 3, 6, and 9 months from those aged for 15 and 18 months, with the latter showing a stronger association with most of the analyzed amino acids. Sensory evaluation revealed that wines aged for 18 months on lees were characterized by aromas of white and citrus fruits, floral and orange blossom notes, honey, butter, toasted bread, and vegetal nuances, along with a straw-colored appearance [46].

More recently, Sawyer S. et al. [47] examined the impact of autolysis and lees aging duration on the aroma profile of Australian sparkling wines produced using traditional grape varieties such as Chardonnay and Pinot Noir via the bottle-fermented method. Two base wines were analyzed at 6, 12, and 24 months post-bottling, with sensory evaluations after 12 and 24 months. Aging time significantly influenced fermentation-derived and oxidative aroma compounds, while the contribution of autolysis products was less pronounced than expected, suggesting that maturation-related compounds play a more dominant role in shaping aroma [47].

Since the aging of lees is a time-consuming process, several studies have explored different strategies to enhance both the quality and efficiency of sparkling wine production. A thorough review by Cravero, M. [7] summarized research conducted in recent years on the application of various techniques to improve the characteristics of sparkling wines [7]. Among these approaches, notable examples include the use of β-glucanases and yeast derivatives (such as autolyzed yeasts and yeast cell walls) to enhance the sensory attributes of Verdejo sparkling wines [48]; the application of ultrasound treatment on yeasts before the second fermentation to facilitate the release of intracellular compounds in Tempranillo wines [49]; and the use of different volumes of lees recovered from the first fermentation to enhance the finesse and complexity of Bombino’s sparkling wines [29].

In agreement with Ruipérez, V. et al. [48], the use of β-glucanases proved to be a promising tool in accelerating the aging of traditional Verdejo sparkling wines, with an increase in the antioxidant activity, but its effects are dependent on the strain used for secondary fermentation [50,51,52]. While yeast derivatives improved the fruity and floral character of sparkling wines, β-glucanases presented a higher yeasty aroma [48,52].

Additionally, Sartor, S. et al. [53] investigated the impact of mannoprotein addition on the chemical composition of Brazilian rosé sparkling wines during lees aging. While mannoproteins had no significant effect on pH or titratable acidity, treated wines exhibited lower volatile acidity, higher concentrations of free and total SO_2_, and increased alcohol content compared to control samples. Mannoproteins also influence color parameters and the composition of individual phenolic compounds. Their positive effects were most pronounced at the end of biological aging, with notable increases in trans-resveratrol, quercetin, catechin, *p*-coumaric acid, and hydroxybenzoic acid [53]. These results are consistent with those from Moyano-Gracia, R. et al. [54], who reported a slight increase in both total acidity and pH over time in red Tempranillo sparkling wines treated with mannoproteins [54].

The effect of accelerated autolysis of yeast on the composition and foaming properties of sparkling wines was also evaluated [55]. The wines were produced using the Parellada white grape variety, and those fermented with the mutant strain IFI473I demonstrated an accelerated release of protein, amino acids, and polysaccharides, which may significantly reduce production times [55].

The research described in this chapter not only offers valuable insights into aging mechanisms but also reflects a growing interest in optimizing quality and wine style diversity, highlighting the need for continued innovation in both analytical techniques and winemaking practices.

### 2.4. Riddling, Disgorgement, and Commercialization

The final stages of traditional sparkling wine production involve *rémuage*, a systematic riddling process that concentrates yeast sediment in the bottle neck, followed by disgorgement, in which the inverted neck is immersed in a glycol bath to freeze the sediment and enable its expulsion. Afterwards, the expedition liquor is added, contributing to the unique sensory profile of the wine [8]. The choice of cork stoppers is critical because they are responsible for the preservation of dissolved CO_2_ and VOCs. Several cork types are commercially available, from microagglomerated stoppers to agglomerated bodies with one to three natural cork discs. Amaro, F. et al. [2] demonstrated that, after 42 months of storage, corks with natural discs preserved ethyl esters, contributing to a fruitier and sweeter aroma of the sparkling wine [2].

Storage conditions also play a significant role in the chemical and sensory evolution of sparkling wines. Factors such as temperature, light exposure, bottle position, and oxygen ingress modulate aroma stability. It has been reported that thermal shifts can influence the formation of furans and reduce terpene content, and pH alterations can increase methional and 1,1,6-trimethyl-1,2-dihydronaphthalene (TDN), considered off-flavors in sparkling wines [56,57,58,59]. A study conducted in Spain revealed that caffeic, trans-coutaric, and *p*-coumaric acids had a direct correlation with browning over time [60].

## 3. Identification and Quantification of Volatile and Semi-Volatile Compounds in Sparkling Wines

Sparkling wine is a complex matrix composed of several compounds that contribute to its unique sensory profile and overall quality. These include volatile aroma compounds, such as esters, alcohols, aldehydes, and terpenes, and non-volatile constituents, such as phenolic acids, organic acids, sugars, proteins, and inorganic acids [44,53,61,62].

A comprehensive characterization of the chemical changes and intricate interactions between compounds during secondary fermentation and aging requires advanced analytical techniques for their accurate identification.

Even though research is mainly focused on extraction and analytical techniques applied to still wines, this chapter provides a comprehensive overview of the methodologies used in the analysis of sparkling wines, discussing their underlying principles, applications, advantages, and limitations.

### 3.1. Sample Preparation

Given the complexity and often low concentrations of some analytes in sparkling wines, sample preparation is a crucial step to accurately identify and quantify these components. These techniques serve to isolate target compounds from the matrix, remove potential interferences, and concentrate the analytes to detectable levels for analytical analysis such as gas chromatography-mass spectrometry (GC/MS) and liquid chromatography-mass spectrometry (LC/MS).

#### 3.1.1. Liquid-Liquid Extraction (LLE)

Liquid–liquid extraction (LLE) is a reference method for the isolation of volatile and semi-volatile compounds in wine due to its efficiency in partitioning analytes between two immiscible liquid phases. This technique allows the extraction of esters, terpenes, alcohols, and organic acids, with solvent polarity playing a critical role in analyte recovery—non-polar solvents favor esters and terpenes, whereas polar solvents enhance the extraction of alcohols and organic acids [63]. Despite its effectiveness, LLE presents significant drawbacks, including the use of environmentally harmful and costly solvents and a lack of automation, resulting in high labor demands [64].

Even though there are not many works regarding sparkling wines and LLE, Voce, S. et al. [65] characterized the volatile compounds of several commercial sparkling wines from the Friuli Venezia Giulia region using three different extraction methods—LLE, solid-phase extraction (SPE), and solid-phase microextraction (SPME). LLE was used for the analysis of non-varietal aromas [65]. In addition, Pérez-Magariño, S. et al. [66] extracted volatile compounds using LLE in a study of white and rosé base wines elaborated from different autochthonous grape varieties in Spain and their evolution to sparkling wines [66].

To minimize solvent consumption, liquid–liquid microextraction (LLME) can be employed as an efficient alternative [67]. This technique involves the use of very small volumes of solvents and samples (typically in the microliter range), allowing for the effective extraction of target compounds with reduced reagent use and shorter processing times. This technique has been successfully applied to the determination of flavor compounds in wine matrices [68,69].

#### 3.1.2. Solid-Phase Extraction (SPE)

In comparison with LLE, solid-phase extraction (SPE) offers a valuable alternative, addressing several limitations of LLE, including the extensive use of organic solvents, multiple procedural steps, longer operation times, increased potential for error, and higher costs [70]. SPE operates on the principle that analytes exhibit a higher affinity for solid adsorbent particles than for the surrounding liquid matrix. As the liquid sample passes through the adsorbent, target compounds are retained and subsequently eluted using an appropriate solvent. In a detailed review, Badawy, M et al. [70] explored both the theoretical and practical aspects of SPE, detailing its various types and their application across diverse real-world matrices [70].

Regarding sparkling wines, SPE has been successfully employed in the isolation of a wide range of volatile compounds. For instance, Slaghenaufi, D. et al. [71] performed an extensive characterization of commercially available Prosecco sparkling wines, using SPE to extract volatile compounds from the samples [71]. Similarly, in sparkling wines from the Friuli Venezia Giulia region, SPE was employed to evaluate both free and bound terpenes and C13-norisoprenoids [65]. Binati, R. et al. [72] used SPE for the identification and posterior quantification of volatile compounds to investigate the contribution of non-Saccharomyces yeast strains (*Lachancea thermotolerans*, *Metschnikowia* spp., and *Starmerella bacillaris*) to the volatile and sensory diversity of wines [72].

SPE was also employed in the isolation of minor and trace volatile compounds in base wines and both free and bound volatile aroma compounds in wines subjected to bentonite and tannin additions during fermentation [73,74,75].

#### 3.1.3. Solid-Phase Microextraction (SPME)

Solid-phase microextraction (SPME) has emerged as one of the most used techniques for the extraction and pre-concentration of volatile and semi-volatile compounds in complex matrices such as sparkling wine. Developed in the 1990s by Pawliszy, SPME revolutionized sample preparation by offering a rapid, solvent-free, and efficient method that integrates the sampling, isolation, and enrichment of analytes in a single step [76,77].

The principle of SPME is based on the partitioning equilibrium between analytes in the sample matrix and a stationary phase coated onto a fused-silica fiber. These coatings are typically composed of polymers or carbon-based adsorbents, such as polydimethylsiloxane (PDMS), divinylbenzene (DVB), or Carboxen (CAR), and can significantly influence extraction efficiency and selectivity [42,61,78]. A comparative analysis of commercial Prosecco wines showed that PDMS/DVB was used for sulfur-containing compounds, while DVB/CAR/PDMS was more effective for extracting terpenoids and norisoprenoids, underscoring the importance of selecting the appropriate fiber to ensure optimal compound extraction [71].

SPME can be employed in two main modes, namely, direct immersion (DI-SPME) and headspace (HS-SPME). In DI-SPME, the fiber is directly immersed in the liquid sample, allowing analytes to diffuse into the coating. Conversely, in HS-SPME, the coated fiber is exposed to the headspace above the sample, where analytes partition between the sample, headspace, and fiber coating. Since the fiber does not make contact with the liquid matrix directly in HS-SPME, the risk of matrix interferences is reduced, and the lifetime of the fiber is significantly prolonged [79]. In the context of sparkling wines, HS-SPME has proven to be a powerful tool for profiling aroma-active compounds such as esters, alcohols, ketones, terpenes, norisoprenoids, and sulfur-containing volatiles, and its application is well-documented in the literature [2,6,15,29,41,42,44,47,49,61,65,80,81,82,83,84,85,86,87].

Despite its many advantages, factors such as temperature, extraction time, and matrix composition can affect HS-SPME efficiency, particularly for low-volatile compounds. Tufariello, M. et al. [88] and Davis, P. et al. [89] demonstrated how matrix effects can impact quantification accuracy and highlighted the importance of calibration strategies and appropriate internal standards [88,89].

Nevertheless, SPME still remains one of the most preferred techniques for the extraction of volatile compounds, due to its simplicity and speed and the fact that it does not require extensive sample preparation. This was demonstrated by Bosch-Fusté, J. et al. [90], who showed it to be a practical alternative to more complex techniques like simultaneous distillation or closed-loop stripping, making it ideal for routine analysis of sparkling wine volatile compounds [90].

#### 3.1.4. Stir Bar Sorptive Extraction (SBSE)

Alternative configurations to traditional SPME have emerged over the years to enhance extraction efficiency and sensitivity in wine analysis. One example is stir bar sorptive extraction (SBSE), which employs a magnetic stir bar, known as a twister, encapsulated in glass and coated with a sorptive phase, typically PDMS or ethylene glycol (EG) [76,79]. In the same way as SPME, this technique relies on the partitioning of analytes between the sample and the sorbent phase. However, it offers significantly improved sensitivity (50–250 times greater) because of the larger capacity of sorptive material in the device (24–126 μL versus 0.6 μL in SPME), which enables greater analyte uptake. This is particularly relevant for trace analysis in complex matrices such as musts and sparkling wines [67,91,92,93].

In addition to its higher sensitivity, SBSE allows for the extraction of analytes from larger sample volumes, increasing its versatility. It has been applied successfully in the quantification of varietal and fermentative volatiles in sparkling wines using approaches such as SBSE coupled with liquid desorption and large volume injection GC–qMS [94], as well as SBSE followed by thermal desorption in the GC-MS system [95,96].

Despite these advantages, SBSE requires thermal desorption units (TDUs), as the device is not compatible with conventional GC split/splitless injectors (only if retro extraction is performed to a suitable solvent) and the range of commercially available coatings is limited to PDMS and EG, which can restrict selectivity and application [79,97].

#### 3.1.5. Thin-Film Solid-Phase Microextraction (TF-SPME)

Following the evolution of sorptive extraction techniques and to address some of the inherent limitations of conventional SPME, thin-film solid-phase microextraction (TF-SPME) has emerged. By employing a planar sorbent-coated surface (carbon mesh or metallic strip), TF-SPME offers a considerably greater surface-to-sample ratio and sorbent volume, resulting in enhanced extraction capacity, faster kinetics, and improved sensitivity for trace-level volatile and semi-volatile compounds [76,98]. Four types of TF-SPME are commercially available: HLB/PDMS (hydrophilic–lipophilic balance), DVB/PDMS, CAR/PDMS, and pure PDMS). The advantage of TF-SPME devices over SBSE is the presence of the solid sorbent, which enhances the extraction of a wide range of analytes.

Even though there are not many studies regarding sparkling wines, in recent years, Marín-San Román, S. et al. [99] optimized TF-SPME conditions for the analysis of volatile compounds in grape musts, evaluating two commercially available coatings—PDMS/CAR and PDMS/DVB. Through a multifactorial design combining variables such as extraction mode, temperature, stirring speed, and duration, it was determined that PDMS/CAR in direct immersion mode (500 rpm, 6 h, 20 °C) yielded the most efficient extraction [99]. Wieczorek, M. [76] demonstrated the potential of HLB-TF-SPME to comprehensively capture volatile profiles in challenging matrices and, Grazioso, T. et al. [98] employed sequential TF-SPME to investigate the aroma profile of sparkling wine, showing that longer extraction times could minimize competitive displacement effects among polar analytes and improve calibration linearity compared to conventional microextraction formats [76,98].

As well as SBSE, TF-SPME represents a significant advancement in sample preparation for volatile compound analysis in sparkling wine, offering improved performance in sensitivity, selectivity, and reproducibility over traditional fiber-based approaches.

#### 3.1.6. QuEChERS

Another method used in sample preparation is QuEChERS (Quick, Easy, Cheap, Effective, Rugged, and Safe). Originally developed for the detection of pesticide residues in fruits and vegetables, this technique has since been adapted to various complex matrices, including wine. Its popularity lies in its simplicity and efficiency, as well as in the minimal solvent consumption and reduced sample handling required [100]. According to the International Organisation of Vine and Wine (OIV-MA-AS323-08) [101], the QuEChERS protocol for pesticide residue analysis in wine involves extraction with acetonitrile, followed by a salting-out partition step using magnesium sulfate and sodium chloride, buffered with citrate salts. The extract is then cleaned up through dispersive solid-phase extraction (d-SPE) using an amino-propyl sorbent and additional magnesium sulfate to remove matrix interferences. To ensure the stability of the extract before instrumental analysis, a small amount of formic acid is added. The final extract can then be analyzed by GC-MS or LC-MS/MS [101].

While there are currently no reports specifically addressing the use of QuEChERS for the extraction of pesticide residues in sparkling wines, the technique has demonstrated promising results in related matrices. For instance, Sykali, D. et al. [100] successfully optimized and validated a QuEChERS-based method for the determination of pesticide residues in grapes, musts, and still wines. Given the similarities in matrix composition, QuEChERS could also be a suitable and effective alternative for sparkling wine analysis, particularly in quality control procedures where multi-residue detection is required [100].

To support the selection of the appropriate sample preparation technique according to the characteristics of volatile compounds and analytical objective, a decision diagram is presented in Figure 3. This figure summarizes the methodologies described in Section 3.1.1, Section 3.1.2, Section 3.1.3, Section 3.1.4, Section 3.1.5 and Section 3.1.6 and illustrates their suitability depending on compound classes, sensitivity, and matrix complexity. Visual guidance is useful for researchers who aim to optimize extraction efficiency.

### 3.2. Analytical Techniques

Traditionally, the analysis of sparkling wines is commonly performed either by using gas chromatography (GC) or liquid chromatography (LC), often followed by mass spectrometry (MS) detection. These approaches offer robust capabilities for the separation and quantification of volatile and non-volatile compounds. In addition to these conventional methodologies, spectroscopic techniques such as ultraviolet–visible (UV–Vis), fluorescence, and nuclear magnetic resonance (NMR) have also been applied to investigate specific classes of compounds, including phenolics, proteins, and metabolites. Table 1 summarizes the most relevant analytical techniques reported in the literature, outlining their main applications in the context of sparkling wine analysis.

#### Alternative Analytical Techniques for the Identification of Target Compounds in Sparkling Wines

Although the most commonly used analytical techniques for the identification of target compounds in sparkling wines have been discussed above, several authors have explored alternative and innovative approaches. Sparkling wine is a complex matrix, characterized by a high concentration of dissolved CO_2_, which necessitates prior degassing before analysis to prevent damage to analytical instrumentation.

Yeast cells can be isolated by centrifugation, as described by Gallardo-Chacón, J. et al. [13]. Alternative methods, such as electrophoresis and PCR (Polymerase Chain Reaction), have also been successfully employed [121]. Tofalo, R. et al. [21] used the MATS method (Microbial Adhesion to Solvents), which involves evaluating cell surface hydrophobicity to characterize microbial populations [21].

In addition to the classical chromatographic methods described above, other practices have been reported, including Size-Exclusion Low-Pressure Chromatography and Molecular Exclusion Chromatography. These techniques allow for the separation of macromolecules such as proteins and polysaccharides based on their size, offering valuable insights into the composition of sparkling wines at different production stages [29,122].

The analysis of proteins in sparkling wine is critical, particularly in relation to foam stability and turbidity formation. Various methodologies have been reported:

Bradford Assay, a colorimetric method for protein quantification [29,48,50,52,123,124,125].

CI-ELLSA (Competitive Indirect Enzyme-Linked Lectin Sorbent Assay), used to quantify specific protein groups [125].

SDS-PAGE (Sodium Dodecyl Sulfate-Polyacrylamide Gel Electrophoresis), a method for separating proteins and glycoproteins based on molecular weight [125].

Size-Exclusion Fast Protein Liquid Chromatography (FPLC), applied by La Gatta, B. et al. [29] and Lambert-Royo, I. et al. [126] for detailed protein profiling.

Folin–Ciocalteu assay, used for determining total phenol and protein content [37,80,112].

Antioxidant capacity is another extensively explored aspect in sparkling wine research. Techniques commonly used include FRAP (Ferric Reducing Antioxidant Power) [37,111,124], DPPH (2,2-diphenyl-1-picrylhydrazyl radical scavenging assay) [52,80,111,124], and HRSA (Hydroxyl Radical Scavenging Activity) [37,52]. For thiol determination, a DTDP (4,4′-dithiodipyridine) method was employed [13].

Regarding sparkling properties, foamability is defined as a quality attribute. Studies have assessed how various production variables influence foam formation, using the Mosalux technique, which quantifies foam height and stability under standardized conditions [17,18,29,66,112,125,127,128,129].

For elemental sparkling wine characterization, Debastiani, R. et al. [36] employed the PIXE (Particle-Induced X-ray Emission) technique to study the elemental transfer between cork stoppers and white sparkling wine over 18 months [36].

In the oenology field, sensory analysis is usually imperative to assess sparkling wine quality. Evaluations are typically conducted under blind testing, conducted by trained male and female panelists. The professionals rate different attributes such as aroma, taste, mouthfeel, bubble, oxidation, and overall balance. This method allows objective comparisons between chemical analyses.

Considering innovative methodologies, Le Menn, N. et al. [130] introduced a novel sensory methodology to assess the aging potential of wine, a concept traditionally described subjectively by professionals. Unlike conventional one-dimensional approaches, this method uses a three-dimensional sensory framework (incorporating time, quality, and potential). The approach, known as “projective categorization,” integrates multiple dependent variables and offers a visual tool for evaluators to project a wine’s developmental trajectory. This investigation was applied to 33 Champagne reserve wines aged from 1 to 29 years, and the method demonstrated strong discriminatory power regarding aging and provided statistical insight into judge consensus and performance [130].

The combination of instrumental and sensory data, along with emerging strategies such as machine learning, is crucial to better understand the volatile aroma compounds that contribute to the complexity of sparkling wine matrices. In recent years, systematic reviews and meta-analyses have helped consolidate the current knowledge on aroma-active compounds in alcoholic beverages, including sparkling wines [131,132]. These studies provide a broader framework for selecting key analytical targets and interpreting the interactions among volatile compounds that shape wine aroma [131,132].

## 4. Conclusions

This review compiles and discusses the influence of different production stages on the aromatic profile of sparkling wines. Each production step significantly influences the wine quality and therefore requires precise technical control. In the context of secondary fermentation, *Saccharomyces cerevisiae* remains the predominant yeast due to its tolerance to harsh conditions and the extensive availability of commercial strains. These strains offer diverse enological traits, such as autolytic capacity, foam production, and flocculation behavior. Recent research has also explored the use of non-*Saccharomyces* species and co-inoculation strategies to enhance aromatic complexity and meet the growing demand for product differentiation and innovation.

The choice of closure during the tirage and the conditions of bottle aging are crucial for the preservation and modulation of volatile compounds. It is during this aging period that the aromatic profile of sparkling wines is developed and refined. The physicochemical changes have been well documented across different sparkling wines.

To understand these transformations, an analytical characterization of sparkling wine analytes is required. While HS-SPME is widely employed for volatile and semi-volatile extraction, innovative techniques such as SBSE and TF-SPME offer promising alternatives with improved sensitivity and fewer limitations. GC/MS and LC/MS allow comprehensive profiling and the quantification of aroma-active compounds.

Future research should focus on the role of cork stoppers during tirage, as they may contribute to the development of distinctive sensory profiles. Moreover, a greater focus on the microbiological dynamics occurring during the early stages of secondary fermentation and within the sealed bottle may provide new insights into sparkling wine maturation and stability.

The insights presented in this review may benefit not only researchers but also winemakers dedicated to the production of sparkling wines. By thoroughly reviewing studies from different stages of the production process, particularly those related to secondary fermentation, this work provides a consolidated framework that can support quality control strategies and informed decision-making. A deeper understanding of the development of volatile organic compounds opens opportunities for optimizing yeast selection during secondary fermentation and closure selection during the tirage phase to achieve specific sensory outcomes and enhance product differentiation in a competitive market. Furthermore, the identification of volatile markers specific to production methods or terroirs may offer valuable tools for the authentication of sparkling wines and the protection of geographical indications, reinforcing traceability and adding commercial value.

## Figures and Tables

**Figure 1 molecules-30-02825-f001:**
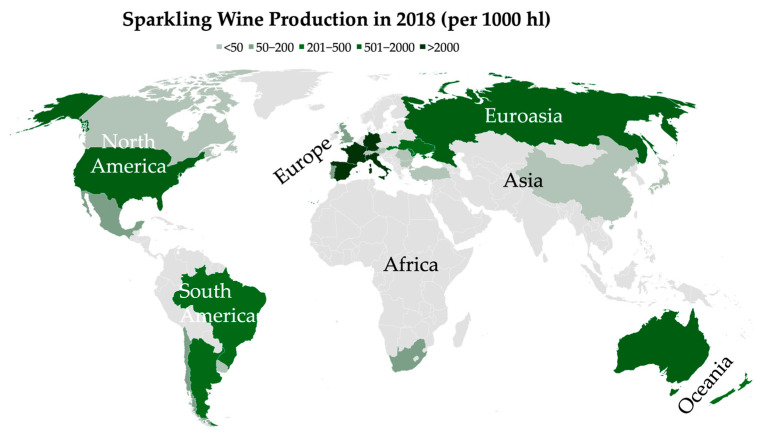
Worldwide sparkling wine production in 2018. Data retrieved from OIV Global Sparkling Wine Market report [4].

**Figure 2 molecules-30-02825-f002:**
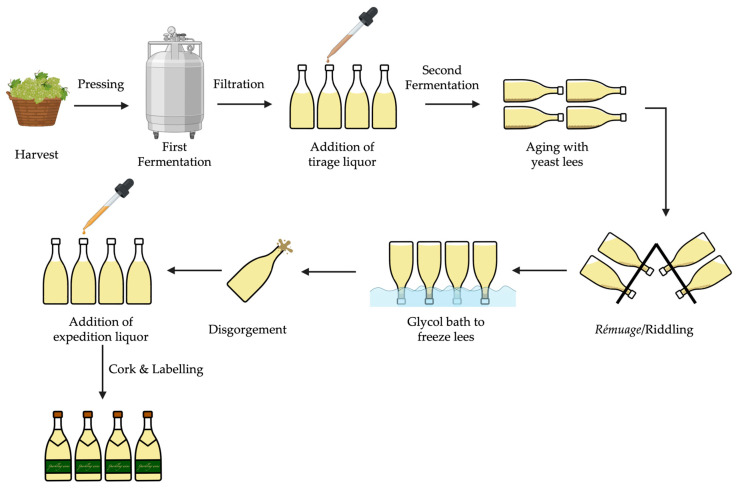
Schematic representation of traditional sparkling wine production.

**Figure 3 molecules-30-02825-f003:**
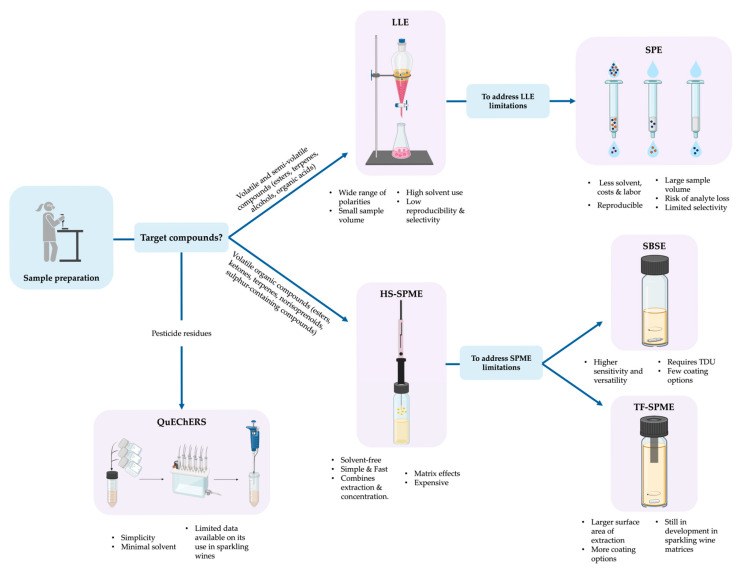
Conceptual decision diagram illustrating the selection of sample preparation techniques for the analysis of volatile and semi-volatile compounds in sparkling wines. The techniques are described in Section 3.1.1, Section 3.1.2, Section 3.1.3, Section 3.1.4, Section 3.1.5 and Section 3.1.6.

**Table 1 molecules-30-02825-t001:** Analytical techniques used in the identification and quantification of target compounds in sparkling wines.

Technique	Detector Type	Analytical Purpose	Reference
Chromatographic Methods			
Gas Chromatography (GC)	Flame Ionization Detector (FID)	Identification of volatile compounds through retention times and quantification using external calibration standards	[11,19,27,78]
	Mass Spectrometry (MS)	Volatile profile of sparkling wines in different research areas: aroma characterization, varietal differences (Cava, Champagne, Italian SW), aging on lees, sulfur compounds, influence of yeast strains and vinification practices.	[2,11,18,19,29,41,42,44,47,49,61,65,66,71,73,78,80,81,84,85,86,88,90,96,98,102,103,104]
	Mass Spectrometry tandem (MS/MS)	Effects of bottle closures, autolysis and aging, and the presence of haloanisoles	[35,47,105]
	Olfactometry (O)	Identification of key aroma compounds; evaluation of antioxidants impact on flavor	[45,73,106,107]
	Pulsed-Flame Photometric Detection (PFPD)	Characterization of volatile sulfur compounds	[73,89]
Multidimensional GC (GCxGC)	Time-of-flight detector (TOF/MS)	VOCs profiling in Italian and Moscatel sparkling wines.	[65,108,109,110]
High-Performance Liquid Chromatography (HPLC) or Ultra-High-Performance Liquid Chromatography (UHPLC) *	Diode Array Detector (DAD)	Used for monitoring organic acids, sugars, glycerol, amino acids, and amines during aging and evaluating the effects of β-glucanases and yeast products on chemical and sensory properties	[46,50,52,60,111,112]
	Fluorescence Detector (FLD)	Monitorization of amino acids and quantification of biogenic amines	[27,113]
	Mass spectrometry (MS)	Analysis of anthocyanins, polysulfides in aged sparkling wines, and chemical profiling of Bombino sparkling wines produced with autochthonous yeast strains	[25,27,29,114]
	Mass spectrometry tandem (MS/MS)	Quantification of indoles, aromatic amino acid metabolites, and lipids in sparkling wines.	[65,115,116]
Spectroscopic Methods			
Proton NuclearMagnetic Resonance (^1^H NMR)	Radiofrequency (RF) detector	Characterization of compounds extracted from cork by wine; analysis of sparkling wines aged with different sugars in the expedition liquor	[117,118]
Fourier Transform Infrared Spectroscopy (FTIR)	Interferometer & IR Detector	Oenological analysis of sparkling wines	[15,18,27]
Raman spectroscopy	Photodiode detector	Elemental composition of sparkling wines treated with mannoproteins	[49]
UV-Visible Spectrophotometry (UV-Vis)	UV-Vis Absorbance Detector	Analysis of color intensity, quantification of polyphenols, hydroxycinnamates, and flavonoids, study of antioxidant activity, and spectrophotometric analysis of phenolic compounds in sparkling wines treated with β-glucanases and mannoproteins	[19,27,50,51,52,53,60,111,119]
Spectrometric Methods			
Inductively Coupled Plasma (ICP)	Multicollector-ICP-Mass spectrometry (MC-ICP-MS)	Sr and Pb isotopic marks applied to the authentication of sparkling wines	[120]
	Mass Spectrometer (ICP-MS) Quadrupole Mass Analyzer	Elemental characterization of musts and wines based on biogenic amines	[113]
	Optical Emission Spectroscopy (OES)	Element composition of sparkling wines treated with mannoproteins, and musts based on biogenic amines	[53,113]

* UHPLC enables faster analyses, better resolution, and greater sensitivity compared to traditional HPLC.

## Data Availability

Not applicable.

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
