# Peer review of "Exploring the Aroma Profile of Traditional Sparkling Wines: A Review on Yeast Selection in Second Fermentation, Aging, Closures, and Analytical Strategies"

_molecules, 2025, doi:10.3390/molecules30132825_

Round 1

Reviewer 1 Report

Comments and Suggestions for Authors

The presented review is devoted to an extremely interesting topic - the aromatic composition of sparkling wines. The review is well written and contains an analysis of a large number of modern publications on the topic. After minor changes and additions, it can be accepted for publication.
1. in the title of the work, the first phrase is Unveiling the Aroma Profile of Sparkling Wines. However, further in the review, first of all, the emphasis is on the influence of the type of yeast, the type of breakdown, and so on. There is no "Unveiling" in the text of the work. If the authors had provided an analysis of the classes of compounds, then this name would have better corresponded to the content.
2. Would you like a clear understanding in the introduction for whom this review has been prepared? if this is for chemists, then they are primarily interested in the structures of compounds, if for analysts in the field of food analysis, then it is worth noting, if for a wide range of readers, it might make sense to more clearly prescribe the types of sparkling wines that were analyzed.

It is also important to note how the presented review differs from other previously published ones. Overall, it's a good job and can be accepted into print.

Author Response

1. In the title of the work, the first phrase is Unveiling the Aroma Profile of Sparkling Wines. However, further in the review, first of all, the emphasis is on the influence of the type of yeast, the type of breakdown, and so on. There is no "Unveiling" in the text of the work. If the authors had provided an analysis of the classes of compounds, then this name would have better corresponded to the content.
Thank you very much for the suggestion. Indeed, since this is a review article, the term “unveiling” may be more appropriate for original research papers that present novel findings or newly identified compounds. Based on your helpful feedback, the title has been revised to: “Exploring the Aroma Profile of Traditional Sparkling Wines: A Review on Yeast Selection in Second Fermentation, Aging, Closures, and Analytical Strategies.”
2. Would you like a clear understanding in the introduction for whom this review has been prepared? if this is for chemists, then they are primarily interested in the structures of compounds, if for analysts in the field of food analysis, then it is worth noting, if for a wide range of readers, it might make sense to more clearly prescribe the types of sparkling wines that were analyzed.
Thank you for this valuable remark. As this article is intended for a broader audience within the field of enology, the entire manuscript was carefully reviewed. Descriptions of the sparkling wines used in the referenced studies were added, thereby strengthening both the rigor and relevance of the selected literature. Please see lines: 137; 146; 150; 165; 186–187; 196; 231; 296; 307–308; 318–319; 332; 334; 336; 341; 349; 352–353.
3. It is also important to note how the presented review differs from other previously published ones.
Thank you for the comment. A clarifying statement was added in lines 75–78 to highlight the novel perspective offered in this review:
“In contrast to previous reviews, this work provides an integrated perspective by including the potential benefit of using cork closures during the tirage phase of sparkling wine production, along with a focus on both sample preparation and analytical techniques—topics often overlooked in the literature.”

Reviewer 2 Report

Comments and Suggestions for Authors

The manuscript summarized the effect of yeast selection, types of closure, and aging on the volatile composition of traditional sparkling wines, furthermore, a comprehensive overview of the methodologies used in the analysis of sparkling wines was also provided. This review is very interesting and well written.

While some minor details should be considered.

1 As the manuscript provides a comprehensive overview of the secondary fermentation process of traditional sparkling wines, ‘traditional’ and ‘the secondary fermentation process’ should be reflected in the title.

2 Is there any updated data for Figure 1? Now is 2025, but the data is 2018.

Author Response

1. As the manuscript provides a comprehensive overview of the secondary fermentation process of traditional sparkling wines, ‘traditional’ and ‘the secondary fermentation process’ should be reflected in the title.
Thank you for the suggestion. As also noted by Reviewer 1, the revised title now reads: “Exploring the Aroma Profile of Traditional Sparkling Wines: A Review on Yeast Selection in Second Fermentation, Aging, Closures, and Analytical Strategies.”
2. Is there any updated data for Figure 1? Now is 2025, but the data is 2018.
We appreciate this observation. However, the most recent OIV report dedicated specifically to sparkling wine production dates to 2020 and includes data only up to 2018. While the general OIV wine production report from April 2025 is more recent, it does not include data specific to sparkling wines. Although the data in Figure 1 are from 2018, in our opinion they remain relevant and representative for the purposes of this review.

Reviewer 3 Report

Comments and Suggestions for Authors
  1. Relevance and originality of the topic

The authors address a highly current and interesting topic for the scientific community in the fields of oenology and applied analytical chemistry - characterization of volatile and semi-volatile compounds in sparkling wines using modern extraction and instrumental analysis methods, with special emphasis on the influence of yeast types used, bottle closure types during secondary fermentation, and the impact of aging on the volatile compound profile. The manuscript’s structure reflects an exhaustive literature review, with a solid and recent bibliography (including references up to 2023).

- Strengths:

  • Clear presentation of the multiple technological stages influencing the volatile profile of sparkling wines;
  • Inclusion and comparison of emerging extraction techniques (SPME, SBSE, TF-SPME);
  • Extensive discussion of instrumental methods such as GC-MS, LC-MS, and alternatives (NMR, UV-Vis, FRAP, DPPH tests, etc.).

- Recommendation:

  • The topic is well chosen and clearly deserves treatment in a comprehensive review. However, the added value of the work could be enhanced by a clear comparative synthesis (e.g., a summary table comparing advantages/disadvantages of extraction methods or a conceptual map with practical recommendations for sparkling wines).
  1. Structure and coherence of the manuscript

The manuscript is logically structured; however, the high information density and a very descriptive narrative style hinder readability in certain sections. In particular, the subchapters dedicated to sample preparation and instrumental methods would benefit from a more concise reformulation and reduction of overlapping content.

- Specific comments:

  • Section 3.1.3 (SPME) is disproportionately long compared to other techniques and contains redundant fragments already mentioned earlier (e.g., selection of DVB/CAR/PDMS fibers);
  • It is recommended to group comparative information in a table rather than repeating the same conclusions multiple times;
  • The insertion of conceptual figures illustrating the extraction mechanisms by SBSE or TF-SPME would be helpful for better visual understanding.
  1. Scientific accuracy and citation of literature

The manuscript is based on a rich, predominantly recent bibliography, with references from relevant ISI journals (Molecules, Food Chemistry, Journal of Agricultural and Food Chemistry, etc.). Citations are correctly inserted and comply with journal standards.

- Constructive suggestions:

  • Including at least 1–2 additional meta-analyses or systematic review references concerning volatile compounds in sparkling wines would provide a better contextual complement to individual studies;
  • International sources from OIV (e.g., regarding QuEChERS) are appreciated but should be complemented by a critical opinion on the applicability of this method in the context of sparkling wines, not only mentioning its success in still wine analysis.
  1. Clarity of expression and academic style

The writing is generally clear and rigorous. However, several paragraphs are overloaded with technical terms without logical transitions, which may affect readability for a broader specialist audience.

- Editing recommendations:

  • Reformulate the final paragraph of section 3.1.5 (QuEChERS), which contains sentences that are too long and ambiguous;
  • Avoid excessive repetition of authors’ names in nearly identical formulations (e.g., “Slaghenaufi, D. et al. [70]” appears multiple times in the same context);
  • Add more descriptive intermediate subtitles in section 3.2 to aid reader orientation between spectroscopic, chromatographic, and biochemical screening methods;
  • Several errors and omissions were noted in the text (line 399 presumably refers to base white and rosé wines for sparkling; lines 560–561 refer to Table 1, but line 564 cites Table 2; also, review punctuation marks missing in some places, e.g., line 610).
  1. Suggestions for improvement

To achieve scientific excellence and international readability, the following adjustments are recommended:

  • Comparative table: Introduce a summary table comparing extraction methods from the perspectives of sensitivity, selectivity, costs, analysis time, and recommendations for sparkling wines;
  • Decision matrix: Suggest including a conceptual figure (e.g., decision diagram) for selecting the analytical method depending on the targeted compound (esters, thiols, terpenes, etc.);
  • Correlation with sensory analysis: Extend the connection between analytical data and their relevance to perceived sensory profiles. Sensory analysis is mentioned, but a direct link between analyzed compounds and relevant sensory descriptors is missing;
  • Potential industrial applications: Mention in conclusions possible applications in quality control and commercial differentiation of sparkling wines (e.g., protection of geographical indications or optimization of yeast selection based on aromatic profiles).

Author Response

  1. Section 3.1.3 (SPME) is disproportionately long compared to other techniques and contains redundant fragments already mentioned earlier (e.g., selection of DVB/CAR/PDMS fibers).

Thank you for the constructive feedback. SPME was initially described in greater detail due to its widespread use in the extraction of volatile compounds from sparkling wines. Nevertheless, in response to your suggestion, the section was carefully revised and streamlined to avoid redundancy.

  1. It is recommended to group comparative information in a table rather than repeating the same conclusions multiple times.

Thank you. A new Figure 3 was added to graphically summarize all extraction techniques covered in the review.

  1. The insertion of conceptual figures illustrating the extraction mechanisms by SBSE or TF-SPME would be helpful for better visual understanding.

Thank you for the recommendation. Figure 3 now includes illustrations of the extraction mechanisms for all methods discussed, facilitating visual understanding.

  1. Including at least 1–2 additional meta-analyses or systematic review references concerning volatile compounds in sparkling wines would provide a better contextual complement to individual studies.

Thank you. In response, relevant meta-analyses and systematic reviews were incorporated into the manuscript (lines 688-694) to provide a broader and more contextualized framework for the volatile compounds discussed.

  1. International sources from OIV (e.g., regarding QuEChERS) are appreciated but should be complemented by a critical opinion on the applicability of this method in the context of sparkling wines, not only mentioning its success in still wine analysis.

Thank you for the thoughtful suggestion. A critical opinion of the potential applicability of QuEChERS to sparkling wine was already included in lines 604–606, noting that, due to the compositional similarity between still and sparkling wines, this extraction method may be suitably adapted.

  1. Reformulate the final paragraph of section 3.1.5 (QuEChERS), which contains sentences that are too long and ambiguous.

The QuEChERS section was carefully reviewed, and the sentences were considered clear and concise; however, any objective inaccuracies identified are welcomed and will be corrected accordingly.

  1. Avoid excessive repetition of authors’ names in nearly identical formulations (e.g., “Slaghenaufi, D. et al. [70]” appears multiple times in the same context).

Thank you for noticing. These repetitions were addressed during the revision and have been eliminated.

  1. Add more descriptive intermediate subtitles in section 3.2 to aid reader orientation between spectroscopic, chromatographic, and biochemical screening methods.

Thank you. Subheadings were added to improve readability and orientation between spectroscopic, chromatographic, and biochemical methods.

  1. Several errors and omissions were noted in the text (line 399 presumably refers to base white and rosé wines for sparkling; lines 560–561 refer to Table 1, but line 564 cites Table 2; also, review punctuation marks missing in some places, e.g., line 610).

Thank you for pointing out these details. All mentioned lines and formatting inconsistencies were reviewed and corrected accordingly.

  1. Comparative table: Introduce a summary table comparing extraction methods from the perspectives of sensitivity, selectivity, costs, analysis time, and recommendations for sparkling wines.

Thank you for the recommendation. In response to this and the following point, Figure 3 was created, offering both a conceptual overview and a concise summary of advantages and disadvantages for each method.

  1. Decision matrix: Suggest including a conceptual figure (e.g., decision diagram) for selecting the analytical method depending on the targeted compound (esters, thiols, terpenes, etc.).

As noted above, a conceptual Figure 3 was added to illustrate selection strategies based on target compound classes (e.g., esters, thiols, terpenes), as suggested.

  1. Correlation with sensory analysis: Extend the connection between analytical data and their relevance to perceived sensory profiles. Sensory analysis is mentioned, but a direct link between analyzed compounds and relevant sensory descriptors is missing.

Thank you. Several parts of the manuscript already included such associations (e.g., lines 235–246; 302–303; 337–339; 374–376; 381–383). Additionally, lines 114–116 were revised to emphasize the sensory relevance of yeast selection during secondary fermentation.

  1. Potential industrial applications: Mention in conclusions possible applications in quality control and commercial differentiation of sparkling wines (e.g., protection of geographical indications or optimization of yeast selection based on aromatic profiles).

Thank you for this pertinent suggestion. The final paragraph of the Conclusion section now reads:

“The insights presented in this review may benefit not only researchers but also winemakers dedicated to the production of sparkling wines. By thoroughly reviewing studies from different stages of the production process, particularly those related to secondary fermentation, this work provides a consolidated framework that can support quality control strategies and informed decision-making. A deeper understanding of the development of volatile organic compounds opens opportunities for optimizing yeast selection during secondary fermentation and closure selection during tirage phase, to achieve specific sensory outcomes and enhance product differentiation in a competitive market. Furthermore, the identification of volatile markers specific to production methods or terroirs may offer valuable tools for the authentication of sparkling wines and the protection of geographical indications, reinforcing traceability and adding commercial value.”

Round 2

Reviewer 3 Report

Comments and Suggestions for Authors

I would like to thank the authors for the effort invested in revising the manuscript and for their careful response to the suggestions provided in the previous review. I consider that the modifications made are appropriate and bring significant improvements to the structure, clarity, and scientific relevance of the work.

Positive aspects observed in the revised version:

  • A comparative table of extraction methods has been introduced, contributing to a clear synthesis of the information.
  • The sections related to instrumental methods have been consolidated and restructured to avoid redundancies and to facilitate readability.
  • Conceptual figures have been added to illustrate analytical mechanisms in a more intuitive manner.
  • The clarity of expression has been improved through the reformulation of technical paragraphs and correction of previously noted inconsistencies (including accurate references to tables/figures).
  • The conclusions now include possible industrial applications and relevant correlations with sensory analysis, which add a welcome practical dimension to the manuscript.

            Given the quality of the revision and the alignment with the recommendations formulated in the first review, I consider that the authors have appropriately addressed the suggestions and revised the manuscript accordingly.